# Comparative Transcriptomics Analysis for Gene Mining and Identification of a Cinnamyl Alcohol Dehydrogenase Involved in Methyleugenol Biosynthesis from *Asarum sieboldii* Miq.

**DOI:** 10.3390/molecules23123184

**Published:** 2018-12-03

**Authors:** Jinjie Liu, Chong Xu, Honglei Zhang, Fawang Liu, Dongming Ma, Zhong Liu

**Affiliations:** 1School of Pharmacy, Shanghai Jiao Tong University, Shanghai 200240, China; LHJW161903@sjtu.edu.cn (J.L.); fawang90@126.com (F.L.); 2Research Center of Chinese Herbal Resource Science and Engineering, Guangzhou University of Chinese Medicine, Guangzhou 510006, China; 15013053420@163.com; 3Jiusan administration of Heilongjiang farms & land reclamation, Harbin 161441, China; ZHL5555@163.com

**Keywords:** *Asarum sieboldii* Miq., ASARI RADIX ET RHIZOMA, transcriptome, cinnamyl alcohol dehydrogenase, methyleugenol biosynthesis pathway

## Abstract

*Asarum sieboldii* Miq., one of the three original plants of TCM ASARI RADIX ET RHIZOMA, is a perennial herb distributed in central and eastern China, the Korean Peninsula, and Japan. Methyleugenol has been considered as the most important constituent of *Asarum* volatile oil, meanwhile asarinin is also employed as the quality control standard of ASARI RADIX ET RHIZOMA in Chinese Pharmacopeia. They both have shown wide range of biological activities. However, little was known about genes involved in biosynthesis pathways of either methyleugenol or asarinin in *Asarum* plants. In the present study, we performed de novo transcriptome analysis of plant tissues (e.g., roots, rhizomes, and leaves) at different developmental stages. The sequence assembly resulted in 311,597 transcripts from these plant materials, among which 925 transcripts participated in ‘secondary metabolism’ with particularly up to 20.22% of them falling into phenylpropanoid biosynthesis pathway. The corresponding enzymes belong to seven families potentially encoding phenylalanine ammonia-lyase (PAL), trans-cinnamate 4-monooxygenase (C4H), *p*-coumarate 3-hydroxylase (C3H), caffeoyl-CoA *O*-methyltransferase (CCoAOMT), cinnamoyl-CoA reductase (CCR), cinnamyl alcohol dehydrogenase (CAD), and eugenol synthase (EGS). Moreover, 5 unigenes of DIR (dirigent protein) and 11 unigenes of CYP719A (719A subfamily of cytochrome P450 oxygenases) were speculated to be involved in asarinin pathway. Of the 15 candidate CADs, four unigenes that possessed high FPKM (fragments per transcript kilobase per million fragments mapped) value in roots were cloned and characterized. Only the recombinant AsCAD5 protein efficiently converted *p*-coumaryl, coniferyl, and sinapyl aldehydes to their corresponding alcohols, which are key intermediates employed not only in biosynthesis of lignin but also in that of methyleugenol and asarinin. qRT-PCR revealed that AsCAD5 had a high expression level in roots at three developmental stages. Our study will provide insight into the potential application of molecular breeding and metabolic engineering for improving the quality of TCM ASARI RADIX ET RHIZOMA.

## 1. Introduction

The genus *Asarum*, containing about 100 species which have conventionally been used as herbal medicines, are distributed in East Asia (65 species), North America (15 species), and Europe (1 species) [1,2]. The Chinese Pharmacopeia has all along enlisted ASARI RADIX ET RHIZOMA, which is stipulated to be dried roots and rhizomes of three different taxa, that is, *Asarum heterotropoides* Fr. Schmidt var. *mandshuricum* (Maxim.) Kitag. (called Bei Xixin in Chinese), *A. sieboldii* Miq. var. *seoulense* Nakai (called Hancheng Xixin in Chinese), or *A. sieboldii* Miq. (called Hua Xixin in Chinese) [3]. *A. heterotropoides* var. *mandshuricum* and *A. sieboldii* var. *seoulense* are both distributed in mountainous areas around Changbaishan within Northeast China. For the similar habitat and overlapped distribution, *A. heterotropoides* var. *mandshuricum* and *A. sieboldii* var. *seoulense* usually tend to be mixed together so that they are collectively called Liao Xixin in Chinese. In contrast, *A. sieboldii* grows in central and eastern China with the Qinling–Dabashan mountainous areas as the center of its distribution, clearly separating it from the two former varieties on the basis of occupation [1].

However, the reports available until now demonstrated that all three taxa have very similar chemical characteristics and biological activities. Both lignans (especially for asarinin and sesamin) and volatile oil (especially for methyleugenol and kakuol) have been regarded as the major active chemicals of ASARI RADIX ET RHIZOMA [4,5,6]. In traditional Chinese medicine, ASARI RADIX ET RHIZOMA has been widely therapeutically used in numbers of formula for a long history, mainly functioning in treatment of toothache, cough, chronic bronchitis, headache, etc. due to its anti-inflammatory, analgesic, and memory enhancing effects [7,8,9,10,11,12].

Asarinin, often found in vascular plants like *Sesamum* spp, is the epimer of Sesamin, a furofuran class lignan. In *Sesamum indicum* cytochrome P450 (SiP450), CYP81Q1, has been reported to catalyze sesamin biosynthesis by forming methylenedioxy bridges [13,14]. Pinoresinol could be formed from two coniferyl alcohol radical substrates by dirigent protein (DIR) [15,16,17,18,19]. In *A. sieboldii*, asarinin and sesamin hold stronger antitumor, antibacterial, and antiallergy effects [20,21,22,23]. However, asarinin biosynthesis pathway in *A. sieboldii* remains unknown.

Methyleugenol, possessing antitussive [24] and antinociceptive effect [25], is one of the most important constituents of *Asarum* volatile oil. Some studies even revealed the possibility of methyleugenol as a leading compound for anorexia treatment [26]. It is worth highlighting that the biosynthesis of methyleugenol shares the common precursor, coniferyl alcohol, with G-lignin. Due to the importance of lignin in the economy, as well as of methyleugenol in food and cosmetic industry, the elucidation of this metabolic pathway has drawn more attention. Very recently, coniferyl alcohol acyl transferase (CAAT) has been cloned and characterized in apple fruit [27]. Furthermore, eugenol synthase genes (EGS) have also been reported in some species such as Rose, Ocimum, and Gymnadenia [28,29,30]. The genes coding *O*-methyltransferases (EOMT) involved in methyleugenol have been isolated and functionally characterized in loquat (*Eriobotrya japonica*), *Ocimum tenuiflorum*, *Clarkia breweri*, sweet basil (*Ocimum basilicum*), and ripe apple fruit [31,32,33,34,35]. Cinnamyl alcohol dehydrogenase (CAD; EC 1.1.1.195), a member of alcohol dehydrogenase, functions in converting hydroxycinnamyl aldehydes to corresponding alcohols, the various direct precursors of lignin [36], among which the coniferyl alcohol also acts as the precursor in methyleugenol boosynthesis. Therefore, the production of coniferyl alcohol and its metabolic flow plays an essential role in the biosynthesis of methyleugenol. Many studies have performed in its structure properties and activity assay in vitro. For example, the nine putative cinnamyl alcohol dehydrogenase genes were reclassified in *Arabidopsis*. They were divided into three types: two (AtCAD4, AtCAD5) had the highest activity and regarded as bona fide CADs; four (AtCAD2, 3, as well as AtCAD7 and 8) had low activity and highest homology to sinapyl alcohol dehydrogenase; the rest (AtCAD6, AtCAD9) were considered functionally redundant in CAD metabolic networks [37]. In recent years, CADs have been characterized in many plants such as poplars, *Oryza sativa*, Rauvolfia, etc. [38,39,40]. Although the entire biosynthesis pathway of methyleugenol has been completely elucidated in sweet basil [32], methyleugenol biosynthesis pathway in *A. sieboldii* remains poorly understood. Thus, it is significant to isolate and characterize AsCADs in *A. sieboldii*, which is involved in coniferyl alcohol production.

In this study, transcriptomic sequencing and analysis from five tissues at different developmental stages from *A. sieboldii* were performed to uncover candidate genes involved in methyleugenol and asarinin biosynthesis pathway, with more attention to the CADs. Of the 15 candidate CADs, 4 unigenes with the higher FPKM (fragments per transcript kilobase per million fragments mapped) in root were cloned and confirmed by qRT-PCR. Of the four recombinant proteins, only AsCAD5 displayed the capability for converting aldehydes (*p*-coumaryl, coniferyl, and sinapyl aldehydes) to their corresponding alcohols in vitro. Enzyme kinetics of three aldehydes was conducted to understand the suitable substrates. All of the results provide a good scenario for pathway elucidation, as well as CADs functions possibly involved in methyleugenol and asarinin pathway. Our study will help in improving the quality of *A. sieboldii* through breeding and metabolic regulations.

## 2. Results

### 2.1. De Novo Assembly and Sequence Annotation

A morphological feature of the intact plant and the representative morphology of root, leaf (including leaf blade and petiole) and rhizome were shown in Figure 1. A total of 89,141,416; 80,100,512; 86,269,700; 88,819,410; and 92,256,930 clean reads were obtained, respectively from different tissues at different developmental stages: root in May (Sample 1), rhizome in May (Sample 2), rhizome in March (Sample 3), rhizome in July (Sample 4) and leaf in May (Sample 5). All the clean reads of five transcriptome sequencing databases were integrated and assembled into 311,597 unigenes. Taking together, 128,934 unigenes (41.38%) were annotated via public databases including NCBI non-redundant protein (Nr), Swiss-Prot protein, Cluster of Orthologous Groups (COG), and the Kyoto Encyclopedia of Genes and Genomes (KEGG). Among them, 9741 unigenes showed significant matches to all four databases. Unigenes that were annotated as unique in various public databases are as follows: 127,167 unigenes in the Nr database, 76,685 unigenes in the Swiss-Prot database, 22,386 unigenes in the COG database, and 35,326 unigenes in the KEGG database (Table 1). Furthermore, about 58.62% of unigenes (182,663) did not show any matches to known genes. They may be considered novel transcripts and specific genes from *A. sieboldii*.

### 2.2. Gene Ontology Classification, COG Classification, and Metabolic Pathway Assignment by KEGG

GO-Term Finder was frequently used to identify Gene Ontology (GO) terms. In total, 157,713 unigenes possessed assignment to one or more gene ontology categories, while 46,782 unigenes were from the cellular component, 61,524 from the biological process, and 49,407 from the molecular function. All unigenes were predicted and classified against the COG database. As a result, 22,386 unigenes were annotated and grouped into 25 COG classifications. Among them, the cluster for metabolic process was 16,611 (10.53%). Some unigenes were assigned to multiple COG classifications; totally 69,969 COG functional annotations were obtained (Appendix A).

The KEGG annotation results showed 35,326 unigenes (11.63%) were obtained and assigned to 369 KEGG pathways, with leading roles in six categories including metabolism, cellular processes, genetic information processing, environmental information processing, organismal systems, and human diseases (Appendix A, Figure 2). Moreover, metabolism occupied highest number of transcripts from *A. sieboldii*, accounting for 26.59% (22,194). Altogether 925 transcripts were involved in biosynthesis of various secondary metabolites. The cluster for ‘phenylpropanoid biosynthesis [PATH: ko00940]’ and ‘terpenoid backbone biosynthesis [PATH: ko00900]’ represented the largest group (20.22%), which was indicated in Figure 3. This information will provide useful guidance for further research on specific pathways and related genes in *A. sieboldii*.

### 2.3. Gene Expression Analysis of AsCADs by qRT-PCR

For the purpose of clarifying the expression of AsCAD genes in different tissues, the relative expression level of AsCADs was evaluated at three stages, with 18S rRNA as reference gene. Before qRT-PCR experiment, we speculated petiole possibly contained lignin to some extent. Thus, leaf blade and petiole have been separated as qRT-PCR templates. The results showed AsCAD5 in roots had higher expression than in other tissues (e.g., leaf blades, petioles, and rhizomes) at each stage. AsCAD showed a similar expression pattern between roots and rhizomes, AsCAD7 showed a similar expression pattern between different tissues. For AsCAD3, the expression level remained low either in roots or in other tissues when compared with AsCAD, AsCAD5, and AsCAD7 (Figure 4).

### 2.4. Identifying Genes Involved in Biosynthesis of Phenylpropanoids Pathway in A. sieboldii

According to the reported methyleugenol biosynthesis pathway, the corresponding enzymes belonged to eight families potentially encoding phenylalanine ammonia-lyase (PAL) (9 unigenes), trans-cinnamate 4-monooxygenase (C4H) (7 unigenes), 4-coumarate-CoA ligase (4CL) (24 unigenes), *p*-coumarate 3-hydroxylase (C3H) (4 unigenes), caffeoyl-CoA *O*-methyltransferase (CCoAOMT) (8 unigenes), cinnamoyl-CoA reductase (CCR) (40 unigenes), cinnamyl alcohol dehydrogenase (CAD) (15 unigenes), and eugenol synthase (EGS) (2 unigenes) (Figure 5). One family potentially encoding dirigent proteins (DIR) (five unigenes) involved in asarinin pathway has also been identified. Plant cytochrome P450 (P450), CYP719A subfamily, can catalyze methylenedioxy bridge-forming reactions in isoquinoline alkaloid-producing plant species [41]. Eleven unigenes of CYP719A were found in *A. sieboldii* and a heatmap based on transcription levels of 512 CYP450s unigenes in the five samples are shown in Appendix A. It is worth noting that 8 of 11 CYP719A had high expression in roots.

### 2.5. Cloning and Phylogenetic Analysis of AsCADs

Of the 15 candidate CADs, four unigenes AsCAD, AsCAD3, AsCAD5, and AsCAD7 (Genbank accession number: KT454711, KY446442, KY446444, and KY446446) with the higher FPKM were cloned. As shown in Figure 6, the alignment of AsCADs protein sequences with the representatives of the CAD family in *Arabidopsis thaliana*, *Artemisia annua*, *Nicotiana tabacum*, and *Zea mays* uncovered that all of the candidates for AsCADs had a highly conserved catalytic zinc binding domain, and AsCAD and AsCAD7 were highly identical with AtCAD5 at all key residues [42]. In phylogeny analysis AsCADs were gathered into three major groups concerning the CAD protein family (Figure 7). Several classification treatments of the CAD family have been proposed based on substrate specificity, expression pattern and gene structure [43,44,45,46,47]. According to them, AsCAD and AsCAD7 nested in Class I, also called ‘bona fide’. This group was highly conserved in terms of substrate specificity, which was represented by AtCAD4 and AtCAD5, enzymes associated with primary lignin synthesis in *Arabidopsis*. AsCAD3 belonged to Class II, whereas AsCAD5 fell into Class III for its relatively high similarity with AtCAD1.

### 2.6. Cloning and Heterologous Expression of AsCADs

To elucidate biochemical functions of AsCADs, we attempted to clone the candidates (AsCAD, AsCAD3, AsCAD5, and AsCAD7) from wild type *A. sieboldii*. The cDNAs were successfully cloned with open reading frame (ORF) of 1071, 1077, 1068, and 1068 bp respectively. Firstly, the ORFs of them were constructed to His-tagged vector (pDEST 17). Unfortunately, all of recombinant AsCAD proteins exhibited a low-level expression. Thus, MBP-tagged vector (pMAL-C5X) was used for recombinant AsCAD proteins production. They were successfully expressed as soluble protein in Rosetta (DE3) cells and purified with Dextrin Beads 6FF (MBP Affinity chromatography). Before enzyme activities were tested, purified proteins were cleaved by Factor Xa protease to remove MBP tag. All of the four CAD proteins exhibited similar molecular masses of ca. 40 kDa on SDS-PAGE (Figure 8).

### 2.7. CAD Activity and Kinetic Parameters of the Recombinant AsCADs

Three substrates, coniferyl aldehyde, sinapyl aldehyde, and *p*-coumaryl aldehyde, were used to evaluate the enzymatic activity. The enzyme assays revealed that only the recombinant protein coded by AsCAD5 had efficient catalytic activity in converting the substrates to corresponding alcohol products (Figure 9A–C). In addition, kinetic analysis was carried out to characterize Km, Vmax, Kcat, and Kcat/Km values for the three substrates. The microplate reader was used to measure three products’ concentrations considering that they had maximum absorbance values at 340 nm. The resulting data showed that the Km value of coniferyl aldehyde was lower than those of sinapyl aldehyde and *p*-coumaryl aldehyde, which supported a catalytic preference for coniferyl aldehyde. The Kcat/Km values for sinapyl aldehyde and coniferyl aldehyde were similar and higher than for *p*-coumaryl aldehyde (Table 2).

## 3. Discussion

Asarinin, a tetrahydrofurofurano lignan, is the quality evaluation standard of *A. sieboldii* in Chinese Pharmacopeia; and sesamin is the epimer of asarinin. Both have been regarded as the main active ingredients from water-extract of *A. sieboldii* roots. Until now, sesamin biosynthesis pathway has been elucidated in *Sesamum indicum*, in which a cytochrome P450, SiCYP81Q1, has been reported to catalyze sesamin biosynthesis by forming two methylenedioxy bridges [13,14]. Although CYP81 in *A. sieboldii* was not found, 11 unigenes of CYP719A subfamily have been found in our transcriptomics data (Figure 5, Appendix A). It has been reported that CYP719A23 from *Podophyllum hexandrum* and CYP719A24 from *Podophyllum peltatum* were capable of converting (−)-matairesinol into (−)-pluviatolide by catalyzing methylenedioxy bridge formation in the biosynthesis of podophyllotoxin, an aryltetralin lignan extensively used for treating cancer [48]. Likewise, the CYP719A subfamily has also been reported in alkaloid biosynthesis in catalyzing the formation of a methylenedioxy bridge. In cases of alkaloid biosynthesis—CYP719A1, CYP719A2, CYP719A3, CYP719A5, CYP719A9, CYP719A13, and CYP719A14—have been reported to be involved in methylenedioxy bridge-formation [41,49,50]. Thus, a proposed pathway was speculated in Figure 3, where CYP719s might participate in the synthesis of sesamin or asarinin by forming two methylenedioxy bridges. Indeed, five DIR unigenes were found in *A. sieboldii*, suggesting two molecules of coniferyl alcohols could be used for formation of pinoresinol. Taking together, the biosynthesis pathways of asarinin in *A. sieboldii* were deduced in the light of transcriptome sequencing data. One possible pathway of asarinin is that two molecules of conifer alcohols were catalyzed by DIRs and CYPs to form sesamin, then transformed to asarinin by epimerization [51]. The second possible pathway of asarinin biosynthesis is that two molecules of conifer alcohols were catalyzed by DIRs to form pinoresinol epimer, then directly transformed to asarinin by CYP719As (Figure 5).

Methyleugenol, eugenol, safrole, and other eugenol derivatives, the natural constituents of many plant essential oils, exist not only in *A. sieboldii* (Aristolochiaceae) but also in plants of other families such as Piperaceae, Labiatae, Myristicaceae, and Lauraceae (trees) [52,53,54,55,56,57,58,59,60,61,62]. Safrole and eugenol are known carcinogens in animals and methyleugenol is a suspected carcinogen [63]. For this harmful effects or uncertain safety of these mentioned chemicals, the utilization and content control of them in food and everyday articles have obtained more and more attention. A decoction procedure, similar to that traditionally used for Chinese herbal preparations, is able to effectively reduce the amount of safrole and methyleugenol in *Asiasari radix* [64]. In our study, two unigenes potentially encoding eugenol synthase (EGS) have been found in the trancriptomes of five tissues. Furthermore, safrole (C_10_H_10_O_2_) could be speculated to be formed from eugenol (C_10_H_12_O_2_) through the formation of the methylenedioxy bridge by CYP719As. As a result, a proposed pathway of safrole biosynthesis was shown in Figure 5. If the EGS, EOMT, and CYP719A functions would be elucidated, reduction of eugenol, safrole, and methyleugenol could be achieved by knockdown or knockout of the corresponding genes.

From pathway shown in Figure 5 we can see that asarinin, sesamin, methyleugenol, eugenol, and safrole share a common precursor coniferyl alcohol, which is generated from coniferyl aldehyde under the catalysis of CAD proteins. The amino acid sequence alignment showed that two key residues for substrate binding (square), two residues for substrate specificity (triangle), and seven key residues for primary lignin synthesis in Figure 6 (yellow background) and Table 3 (circle) were totally different between AsCAD5, AtCAD5, and AaCAD, among which the latter two CADs have been demonstrated for bona fide cinnamyl alcohol dehydrogenase [37,65]. The amino acids discrepancy of AsCAD5 with AtCAD5 and AaCAD might be one of the reasons of clustering, where AsCAD5 is close to AtCAD1, and belongs to Class III but distant from Class I (e.g., AtCAD4 and AtCAD5) (Figure 7). Although AsCAD5 amino acids and clustering differs from AtCAD5, AsCAD5 exhibited the bona fide cinnamyl alcohol dehydrogenase activity similarly with AtCAD5. Furthermore, AsCAD5 displayed its substrate preference in the order coniferaldehyde > sinapaldehyde > *p*-coumaraldehyde, with a similar catalytic efficiency of coniferaldehyde and sinapaldehyde, but higher than that of *p*-coumaraldehyde, indicating AsCAD5 preferred coniferaldehyde as substrate to form cinnamyl alcohol. qRT-PCR showed AsCAD5 remained highly expressed in active compounds-rich roots in an ascending trend during the engaged developmental stages, namely, pre-anthesis, anthesis, and post-anthesis; but in another main medicinal tissue, rhizome, the expression level of AsCAD5 appeared highest at anthesis and then sharply decreased at post-anthesis. Considering the contents of either volatile oil or methyleugenol and asarinin reached the highest level at anthesis, it was suggested that AsCAD5 probably played an important role in biosynthesis of methyleugenol and asarinin.

In conclusion, our transcriptomics data from roots, rhizomes, and leaves revealed candidate genes in biosynthesis of methyleugenol, asarinnin, and safrole. Moreover, a cinnamyl alcohol dehydrogenase AsCAD5 was demonstrated to be responsible for the formation of coniferyl alcohol, the common precursor of methyleugenol and G-ligin.

## 4. Materials and Methods

### 4.1. Plant Materials

Three-year-old seedlings of *A. sieboldii* were collected from its cultivation base in Ningqiang County, Shaanxi Province, China. cDNA libraries were prepared from different tissues at different developmental stages: root in May (Sample 1), rhizome in May (Sample 2), rhizome in March (Sample 3), rhizome in July (Sample 4), and leaf (leaf blade and petiole, used as a whole or separately) in May (Sample 5). All the fresh samples were frozen at −80 °C and used to extract total RNA.

### 4.2. Illumina Sequencing for Transcriptome Analysis

Total RNA of each sample was extracted according to manufacturer’s protocol of RNeasy^®^ Plant Mini Kit (Tiangen, Beijing, China), then quantified and qualified by Agilent 2100 Bioanalyzer (Agilent Technologies, PaloAlto, CA, USA), NanoDrop (Thermo Fisher Scientific Inc., Waltham, MA, USA) and 1% agrose gel. 1 µg of total RNA with RIN value above 7 was used for sequencing on the Illumina HiSeq^TM^ 4000 sequencing platform. All raw transcriptome data were deposited in the GenBank Short Read Archive. The accession numbers were SRR6838612, SRR6838611, SRR6838610, SRR6838609, and SRR6838613 for *Asarum*-1 to *Asarum*-5, respectively.

### 4.3. De Novo Assembly and Functional Annotation Analysis

In order to remove technical sequences, pass filter data of Fasta format were processed by Cutadapt version 1.9.1 (Dortmund, Germany) to be high quality clean data. With paired-end reads, contigs can be identified from the same transcript and the distance between these contigs can be estimated. Then they were assembled by Trinity [66] to get the unigene sequence file. The generated unigenes were annotated based on the Nr, Swiss-Prot protein, COG, and KEGG public databases. GO-Term Finder was used to annotate a list of enriched genes with a significant p-value less than 0.05. To elucidate the active biochemical pathways in *A. sieboldii*, unigenes were compared against the KEGG using BLASTx with an e-value < 1 × 10^−10^ and the corresponding pathways were established.

### 4.4. Quantitative Real-Time PCR Validation

To investigate expression profiles of AsCADs in different tissues at three developmental stages (collected in March, May and July, representing pre-anthesis, anthesis and post-anthesis accordingly), total RNA was extracted from leaf blades, rhizomes, roots, and petioles and the cDNA was reversely transcribed from 1 µg of total RNA using a FastQuantRT Kit (with gDNase) purchased from Tiangen. Quantitative real-time PCR was performed using 0.5 µL of cDNA product and each primer of 10 µL in a 20 µL reaction volume. The primers for qRT-PCR were designed based on the cDNA sequence (Appendix A). 18S rRNA was used as reference gene.

### 4.5. Identifying Genes Related to Biosynthesis of Methyleugenol and Asarinin in A. sieboldii

KEGG pathway annotation is helpful for predicting potential genes and their functions at a whole transcriptome level. By searching the gene name or the EC number, we screened the key enzymes coding sequences involved in the pathway and determined FPKM values for the transcripts. *A. sieboldii* CAD sequences were subjected to homology search using BLASTp against NR database. Heatmap of CYP450s was plotted using the OmicShare tools, a free online platform for data analysis (www.omicshare.com/tools).

### 4.6. Multiple Sequence Alignments and Phylogenetic Analysis of AsCADs

Deduced protein sequences of AsCADs and functional CADs identified from other plant species were retrieved from the NCBI (https://blast.ncbi.nlm.nih.gov) database. These amino acid sequences were engaged in multiple alignment through Clustal Omega (http://www.ebi.ac.uk/Tools/msa/ clustalo), following a phylogeny analysis. 51 CAD proteins were taken into the phylogeny analysis (Appendix A), which was treated accordingly with the MUSCLE, PhyML (http://phylogeny.lirmm.fr/), and TreeDyn 198.3 (via www.phylogeny.fr) softwares (Information Génomique et Structurale, Marseille, France) [67,68].

### 4.7. Cloning of AsCADs

PCR was performed using PrimeSTAR Max DNA Polymerase (Takara, China), accompanying with a touch-down method: initial denaturation at 94 °C for 5 min, following 10 cycles of denaturation (94 °C, 1 min), annealing (1 min) with the temperature decreasing from 65 to 55 °C during these 10 cycles, and then extension at 72 °C (1 min); subsequently, 25 cycles (1 min each) of denaturation (94 °C), annealing (55 °C) and extension (72 °C); and finally extension at 72 °C for 7 min. The specific primers for AsCAD genes were designed according to the screened sequences and provided in Appendix A. The amplification products were analyzed and the amplified fragments of interest in each case were further picked up and purified with the QIAquick gel extraction kit (Qiagen, Hilden, Germany). Then the purified fragments were ligated into pLB cloning vector (Tiangen) and transformed into *E. coli* DH5α cells. The sequence alignments for each gene were performed using DNAMAN (Lynnon Biosoft Company, San Ramon, CA, USA).

### 4.8. Heterogeneous Expression of Recombinant AsCADs

After confirmation of the cDNA sequence for each CAD homologue, the ORF for each gene was individually cloned in pMAL-C5X between restriction sites of *Nde*I and *Eco*RI using In-Fusion Cloning method. The primers were listed in Appendix A. The positive constructs were then transformed into the competent Rosetta (DE3) cells for expression. A single colony of each transformed *E. coli* cell line was first incubated at 37 °C in LB medium containing carbenicillin (25 mg/L) and chloramphenicol (17 mg/L). When OD_600_ reached 0.4–0.6, 0.1 mM isopropyl-β-d-1-thiogalactopyranoside (IPTG) was added into the medium at 16 °C to induce the protein. The cells were allowed to grow for 20 h and finally harvested by centrifugation at 11,500× *g* for 15 min, with the pellets frozen and stored at −80 °C. Frozen cell pellets for each CAD homologue were individually resuspended in binding buffer (20 mL, pH 7.4) containing Tris-HCl (20 mM), NaCl (200 mM), EDTA (1 mM), and DTT (1 mM) and then lysed by sonication for 10 min on ice (3 s pulse at 30W). Subsequently the cell debris for each CAD homologue was removed by centrifugation at 6000× *g* for 20 min at 4 °C.

All purifications were individually performed at room temperature by using a Dextrin beads 6FF (Smart Lifesciences, Changzhou, China) column (25 mL), previously equilibrated in binding buffer. Samples of each of the enzyme preparations were then applied to the affinity column, in which the target proteins would be held by the tags. The unbound proteins would be removed through washing with binding buffer. Target proteins were individually eluted with elution buffer (20 mM maltose in binding buffer), resulting in 1-mL fraction collection of individual sample. For each CAD homologue, aliquots (15 µL) of every other fraction were subjected to SDS-PAGE with gradient (5% acrylamide) gels; proteins were visualized by eStain L1 Protein Staining System (Genscript, Piscataway, NJ, USA). Fractions containing each MBP-tagged CAD (molecular mass ca. 80 kDa) were combined and dialyzed in PD-10 Desalting Columns (GE Healthcare, Uppsala, Sweden). Considering the molecular mass of MBP tag was similar with AsCADs, proteins after removal of MBP tag were used for enzyme assay by Factor Xa protease (New England Biolabs, Hertfordshire, UK).

### 4.9. CAD Activity Assay and Determination of Kinetic Parameters

Three lignin substrates—coniferyl aldehyde, sinapyl aldehyde, and *p*-coumaryl aldehyde (Sigma-Aldrich, St. Louis, MO, USA)—were used to examine the catalytic activity of the recombinant AsCADs. The reactions were initiated by adding 2 µg of the purified protein to a 300 µL reaction mixture containing 0.1 mM substrate, 50 mM Tris-HCl (pH 7.5), 2 mM DTT, and 0.5 mM NADPH as cofactor. The reactions were allowed to proceed for 30 min at 30 °C, stopped by adding 15 µL glacial acetic acid. After mixed with 300 µL methanol and then centrifuged at 6000× *g* for 10 min, the resulting supernatant was performed high performance liquid chromatography (HPLC) analysis with Waters 2695 equipped with sharpsil-U C18 reverse column (250 × 4.6 mm, 5 µm). Methanol-water (60:40, *v*/*v*) was employed as the mobile system, at the flow rate of 0.8 mL/min. The detection wavelengths were 260 nm and 340 nm, and the injected volume was 5 µL. The AsCADs were further characterized the kinetic parameters with spectrophotometric method, employing each reaction in total 200 µL of mixture in 96-well microplates and possessions of enzymatic reaction data through Epoch Microplate reader (BioTek instruments Inc., Winooski, VT, USA). For each substrate a serial of concentrations was subjected to test, in which all other components of the reaction mixture were kept the same as described above. Extinction coefficients for both aldehyde and NADPH were utilized to calculate the relative contribution of each to the 340-nm signal [69]. The resulting data were analyzed using GraphPad Prism 7 software (San Diego, CA, USA) to determine the Km and Vmax values.

## Figures and Tables

**Figure 1 molecules-23-03184-f001:**
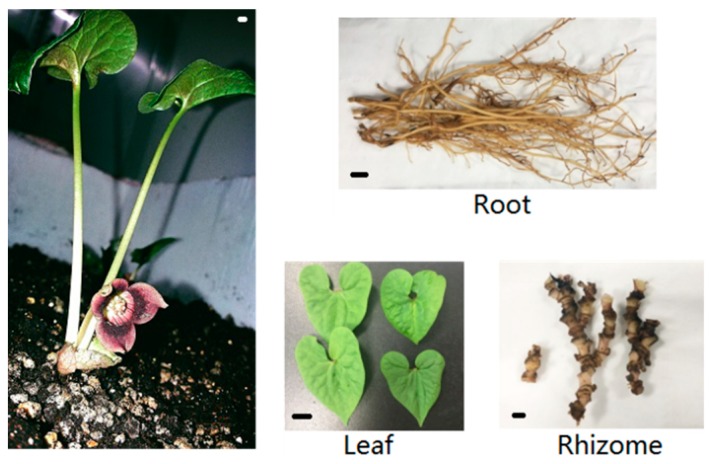
Representative materials diagrams. Scale bars = 1 cm.

**Figure 2 molecules-23-03184-f002:**
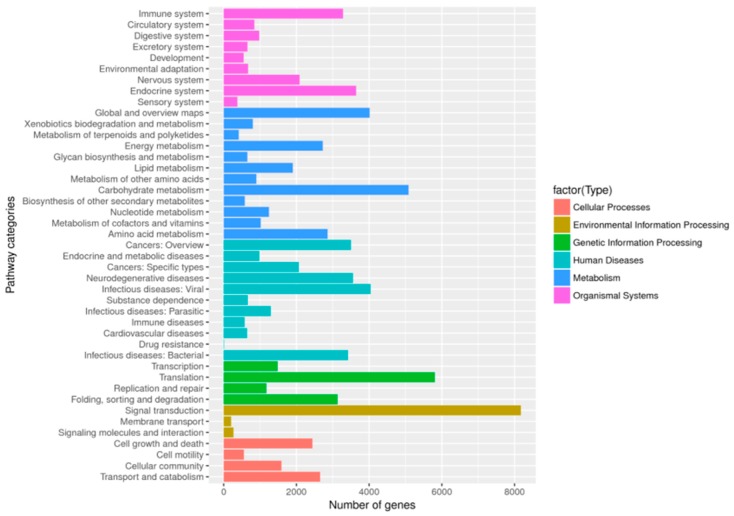
KEGG annotation and classification of unigenes.

**Figure 3 molecules-23-03184-f003:**
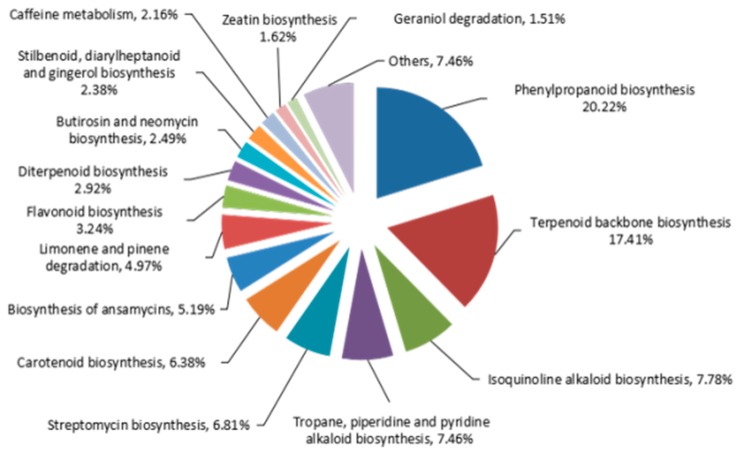
KEGG classification based on secondary metabolism categories. Others containing: tetracycline biosynthesis; novobiocin biosynthesis; brassinosteroid biosynthesis; sesquiterpenoid and triterpenoid biosynthesis; glucosinolate biosynthesis; biosynthesis of siderophore group nonribosomal peptides; betalain biosynthesis; monoterpenoid biosynthesis; polyketide sugar unit biosynthesis; indole alkaloid biosynthesis; flavone and flavonol biosynthesis.

**Figure 4 molecules-23-03184-f004:**
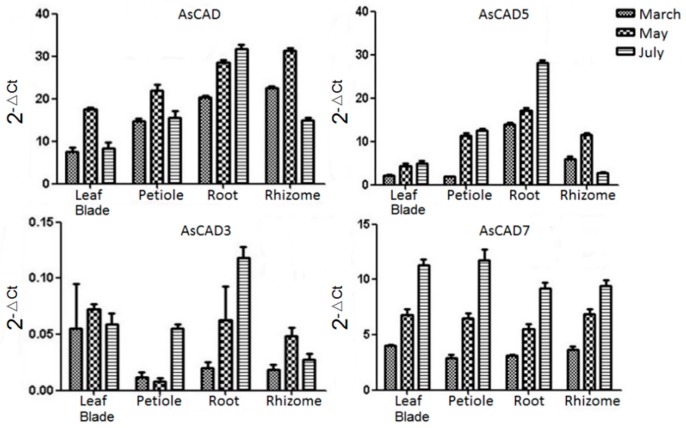
Analysis of AsCADs relative expression in leaf blades, rhizomes, roots, and petioles at three developmental stages (collected in March, May, and July) by real-time quantitative PCR. 18S rRNA was used as reference gene.

**Figure 5 molecules-23-03184-f005:**
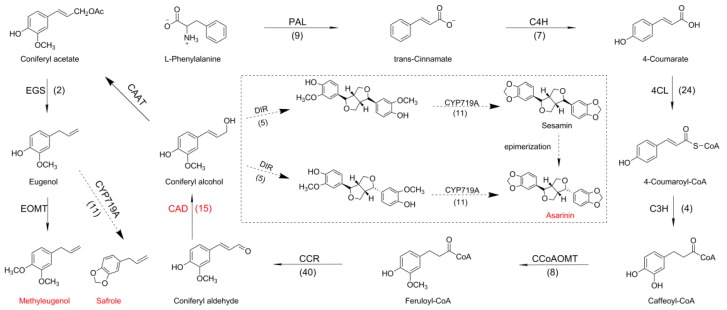
A proposed pathway for methyleugenol and asarinin biosynthesis in *A. sieboldii*. Abbreviations: PAL, phenylalanine ammonia-lyase; C4H, trans-cinnamate 4-monooxygenase; 4CL, 4-coumarate-CoA ligase; C3H, *p*-coumarate 3-hydroxylase; CCoAOMT, caffeoyl-CoA *O*-methyltransferase; CCR, cinnamoyl-CoA reductase; CAD, cinnamyl alcohol dehydrogenase; CAAT, coniferyl alcohol acyl transferase; EGS, eugenol synthase; EOMT, eugenol *O*-methyltransferase; DIR, dirigent proteins; CYP719As belong to cytochrome P450 family. Bracketed numbers represent numbers of transcripts respectively.

**Figure 6 molecules-23-03184-f006:**
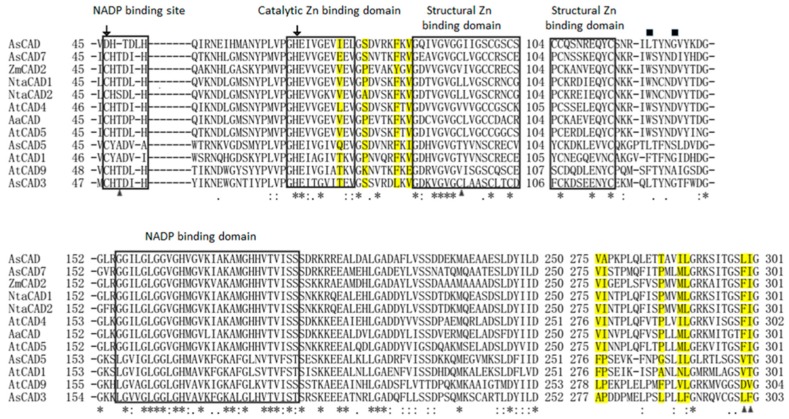
Alignment of putative AsCAD proteins with eight known CAD proteins from other plants. *Arabidopsis thaliana* (AtCAD1, AtCAD4, AtCAD5, AtCAD9), *Nicotiana tabacum* (NtaCAD1, NtaCAD2), *Artermisia annua* (AaCAD), and *Zea mays* (ZmCAD2) (alignment—Clustal Omega software 1.2.2). Arrow: residues common for Zn catalytic activity and substrate specificity; triangle: residues responsible for substrate specificity; square: key residues for substrate binding; asterisk: a single, fully conserved residue; colon: conservation between groups of strongly similar properties; period: conservation between groups of weakly similar properties; yellow mark: key residues for primary lignin synthesis.

**Figure 7 molecules-23-03184-f007:**
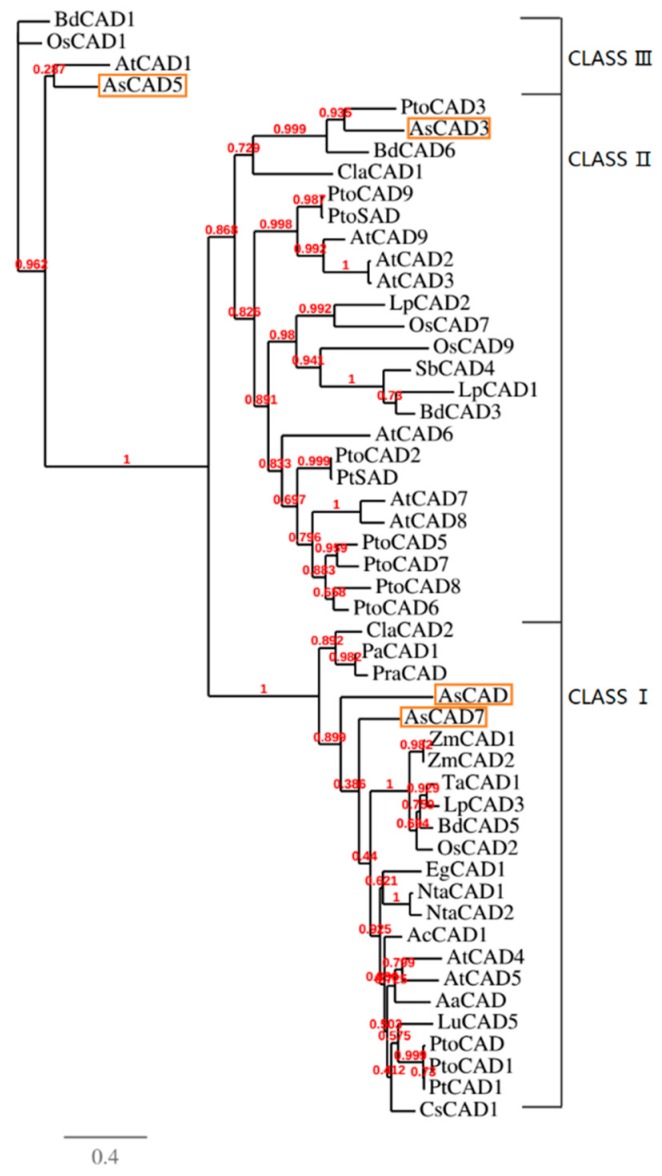
A phylogenetic relationship of AsCADs with CADs from other plant species. The phylogeny analysis was done on a set of 51 CAD proteins, analyzed using PhyML 3.0 (software based on the maximum-likelihood principle) and visualized using TreeDyn software.

**Figure 8 molecules-23-03184-f008:**
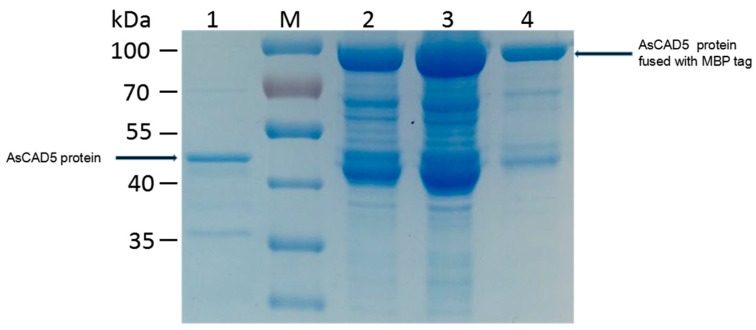
Expression and purification of AsCAD5 recombinant proteins in *E. coli* Rosetta (DE3) harboring pMAL-C5X-AsCAD. M, marker; 1, purified proteins after removal of MBP tag; 2, soluble proteins purified in MBP binding buffer with maltose elution; 3, total proteins induced by IPTG; 4, purified proteins dialyzed in desalting columns.

**Figure 9 molecules-23-03184-f009:**
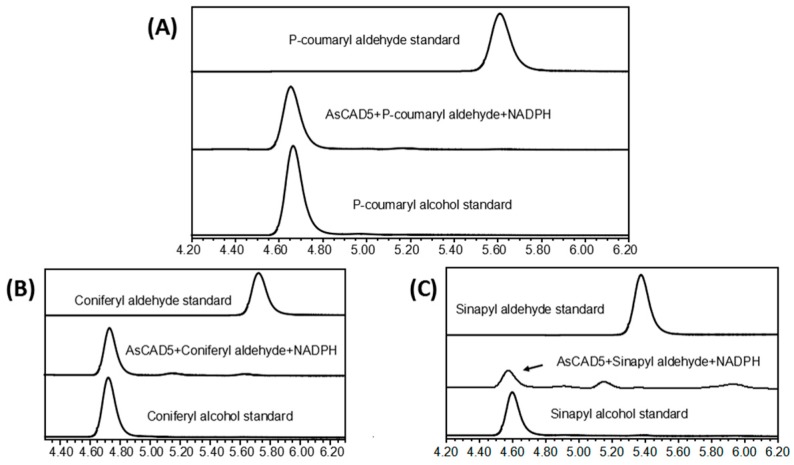
Characterization of enzymatic activity with HPLC analysis and kinetic features: (**A**) profiles of HPLC show conversion of *p*-coumaryl aldehyde to *p*-coumaryl alcohol; (**B**) HPLC profiles show conversion of coniferyl aldehyde to coniferyl alcohol; (**C**) profiles of HPLC show conversion of sinapyl aldehyde to sinapyl alcohol.

**Table 1 molecules-23-03184-t001:** Summary of number of unigenes annotated in *Asarum* as compared to public database

Public Database	Number of Unigenes Annotated in *Asarum sieboldii*
Nr	127,167
Swiss-Prot	76,685
COG	22,386
KEGG	35,326

Note: Total number of annotated unigenes: 128,934; Rate of total annotation: 41.38%.

**Table 2 molecules-23-03184-t002:** Kinetic parameters of the purified recombinant AsCAD5 with three substrates

Enzyme Kinetic Parameters	Coniferyl Aldehyde	Sinapyl Aldehyde	*P*-coumaryl Aldehyde
Km (µM)	27.32 ± 6.186	35.02 ± 8.043	45.52 ± 8.888
Vmax (nmol s^−1^ mg protein^−1^)	342 ± 26.29	448.5 ± 50.54	383.8 ± 31.63
Kcat (s^−1^)	221.7	290.8	248.8
Kcat/Km (µM^−1^ s^−1^)	8.115	8.304	5.466

**Table 3 molecules-23-03184-t003:** Key amino acids diversities among AsCAD5 (*Asarum sieboldii*), AaCAD (*Artemisia annua*), and AtCAD5 (*Arabidopsis thaliana*)

	▲ 48	● 75	● 86	▲ 94	■ 118	■ 122	● 275	● 276	● 285	● 299	● 300
AaCAD	T	V	V	C	W	D	V	I	P	F	I
AtCAD5	T	V	V	C	W	D	V	I	P	F	I
AsCAD5	A	Q	I	T	119T	123S	276F	277P	G	V	T

Triangle: residues responsible for substrate specificity; Square: key residues for substrate binding; Circle: key residues for primary lignin synthesis.

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
