# Peer review of "Comparative Transcriptomics Analysis for Gene Mining and Identification of a Cinnamyl Alcohol Dehydrogenase Involved in Methyleugenol Biosynthesis from Asarum sieboldii Miq."

_molecules, 2018, doi:10.3390/molecules23123184_

Reviewer 1 Report

Authors identified and characterized an enzyme involved in the biosynthesis of methyleugenol, the most important volatile oil in a herb, Asarum sieboldii, used in medicine, by molecular techniques. The study is well done and the manuscript is generally well written. It needs only minor improvement.

Specific comments:

Abstract, line 18: insert "of" biological activities

line 27: insert "and" eugenoly synthase

lines 28/29: explain abbreviations used (DIR etc.)

Introduction line 43: "1 species" is correct

line 87: et al.

Results line 100 and others: use a comma in numbers with more than three digits

line 150 and others: either use lowercase or uppercase letters in names of enzymes

line 168 and others: use a uniform style of writing in subtitles (lowercase or uppercase letters)

line 174: you probably mean "identical"

line 185: Z. mays is the correct name

Legend to Fig. 5: explain abbreviations used for the CAD proteins in the figure

Figure 6: indicate the CAD proteins in the figure, e.g. with an arrow

Discussion, line 233: "was" not found

Materials and Methods, line 318: the use of petioles is not mentioned in 4.1. (plant materials)

line 319: (with...

line 357 and others: give g values in centrifugation steps and not rpm

line 380: give details for that centrifugation step

line 383: insert a space between number and the unit.

The References have to be carefully checked: use a uniform style of writing with either lowercase or uppercase letters in the titles and in journal names; give all species names in italics.

Author Response

Dear Professor,

Thank you for your review and valuable comments on the manuscript “Comparative Transcriptomics Analysis for Gene Mining and Identification of a Cinnamyl alcohol dehydrogenase Involved in Methyleugenol biosynthesis from Asarum sieboldii Miq.”. We have made revisions according to your suggestion as follows:

1. Abstract, line 18: insert "of" biological activities

The "of" has been inserted.

2. line 27: insert "and" eugenoly synthase

The "and" has been inserted.

3. lines 28/29: explain abbreviations used (DIR etc.)

The abbreviations explanations have been added. Exactly, DIR is abbreviation of dirigent protein, and FPKM is that of fragments per transcript kilobase per million fragments mapped.

4. Introduction line 43: "1 species" is correct

It has been corrected.

5. line 87: et al.

  "et al " has been changed to the correct form "et al. ".

6. Results line 100 and others: use a comma in numbers with more than three digits

  They have been added throughout the manuscript. .

7. line 150 and others: either use lowercase or uppercase letters in names of enzymes

They have been uniformed to use lowercase letters in names of enzymes.

8. line 168 and others: use a uniform style of writing in subtitles (lowercase or uppercase letters)

They have been used in uniform style.

9. line 174: you probably mean "identical"

It has been corrected.

10. line 185: Z. mays is the correct name

It has been used the correct form "Z. mays ".

11. Legend to Fig. 5: explain abbreviations used for the CAD proteins in the figure

We have given the details of the CAD proteins in Table S4. Considering convenience for your check, we also list the CAD proteins here:

AtCAD1, AtCAD2, AtCAD3, AtCAD4, AtCAD5, AtCAD6, AtCAD7, AtCAD8, AtCAD9: CADs of Arabidopsis thaliana

AcCAD1: CAD of Aralia cordata

AaCAD: CAD of Artemisia annua

AsCAD, AsCAD3, AsCAD5, AsCAD7: CADs of Asarum sieboldii

BdCAD1, BdCAD3, BdCAD5, BdCAD6: CADs of Brachypodium distachyon

CsCAD1: CAD of Camellia sinensis

ClaCAD1, ClaCAD2: CADs of Cunninghamia lanceolata

EgCAD1: CAD of Eucalyptus globulus

LuCAD1A: CAD of Linum usitatissimum

LpCAD1, LpCAD2, LpCAD3: CADs of Lolium perenne

NtaCAD1, NtaCAD2: CADs of Nicotiana tabacum        

OsCAD1, OsCAD2, OsCAD7, OsCAD: CADs of Oryza sativa

PaCAD1: CAD of Picea abies

PraCAD: CAD of Pinus radiate

PtoCAD, PtoCAD1, PtoCAD8, PtoCAD7, PtoCAD5, PtoCAD3, PtoCAD2, PtoCAD6, PtoCAD9 , PtoSAD: CADs of Populus tomentosa

PtCAD1, PtSAD: CADs of Populus tremuloides

SbCAD4: CAD of Sorghum bicolor

TaCAD1: CAD of Triticum aestivum

ZmCAD1, ZmCAD2: CADs of Zea mays

12. Figure 6: indicate the CAD proteins in the figure, e.g. with an arrow

We have added arrows to indicate AsCAD5 protein and AsCAD5 protein fused with MBP tag.

13. Discussion, line 233: "was" not found

It has been corrected.

14. Materials and Methods, line 318: the use of petioles is not mentioned in 4.1. (plant materials)

For this aspect, it is the reason that the materials used in experiment have not been described clearly. Actually, the tissues employed in the study included all the nutritional organs, i.e., root, rhizome and leaf. In transcriptomic sequencing, the three organs which were at anthesis (May) were used as experiment materials, representing different tissues of the plant. The leaf included leaf blade and petiole. Furthermore, considering the rhizome being the main medicinal part, it was taken as representative of various developmental stages of the plant so that rhizomes at pre-anthesis and post-anthesis (March and July, respectively) were also employed in the sequencing performance. So, there were totally five experiment tissues, with leaf including leaf blade and petiole. (4.1 Plant Materials)

In contrary, what have been used in qRT-PCR analysis were actually also the three nutritional organs, i.e., root, rhizome and leaf. But the organ “leaf” was further separated into leaf blade and petiole. We have revised it to be “leaf blade” instead of “leaf” in this part. (4.4 Quantitative Real-Time PCR Validation)

15. line 319: (with...

 It has been transformed into lowercase letter.

16. line 357 and others: give g values in centrifugation steps and not rpm

The unit has been transformed.

17. line 380: give details for that centrifugation step

The “at 6,000 ×g for 10 min” has been added after “centrifuged”.

18. line 383: insert a space between number and the unit.

The spaces have been inserted.

19. The References have to be carefully checked: use a uniform style of writing with either lowercase or uppercase letters in the titles and in journal names; give all species names in italics.

The references have been carefully checked. Only the first letter of each title is capitalized except for the professional items. All species names involved in references have been uniformed in italics.

Thank you and best regards.

Yours sincerely,

Zhong Liu, Dongming Ma, Jinjie Liu

Reviewer 2 Report

Review of molecules-3597443

 Comparative Transcriptomics Analysis for Gene Mining and Identification of a Cinnamyl alcohol dehydrogenase Involved in Methyleugenol biosynthesis from Asarum sieboldii Miq.  

Jinjie Liu, Chong Xu, Honglei Zhang, Fawang Liu, Dongming Ma and Zhong Liu

The authors sequenced the transcriptomes of roots, rhizomes and leaves of the medicinal plant Asarum sieboldii in order to identify genes involved in the biosynthesis of methyleugenol and asarinin. They identified 9 families of genes potentially encoding enzymes catalyzing each step of the biosynthetic pathways for these compounds. They used bioinformatics tools to identify 15 Asarum sieboldii genes as Cinnamyl alcohol dehydrogenase (CAD) genes based on similarity to previously-identified CAD genes in other plants.  They then cloned the coding sequences of the four most highly expressed CAD genes, AsCAD and AsCADs 3,5, and 7 and fused them to the MBP tag to aid their purification.   The encoded proteins were then expressed in E. coli, the recombinant proteins were purified and their catalytic activity against a panel of substrates was tested.  Only AsCAD5 showed activity toward p-coumaryl, coniferyl, and sinapyl aldehydes. AsCAD5 was shown by qRT-PCR to be expressed at high levels in roots at three developmental stages.

The authors have compiled a useful dataset. The work seems to be performed competently, and the analysis is appropriate. It would be helpful to explain in the introduction and in their results why they chose to focus on CAD genes, rather than waiting until lines 269 and 270 of the discussion.  It also seems that there is excessive speculation about the pathway for asarinin synthesis in the discussion since there is no supporting biochemical evidence from their study.

Otherwise I think that the report is well-organized and provides useful information.

The English is adequate and mostly understandable, but there are numerous mistakes that should be corrected.  I have listed a few below, but there are many throughout the ms, starting in the abstract.

Lines 50-52: “Contrarily, A. sieboldii takes place in central and eastern China with the Qinling-Dabashan mountainous areas as distribution center, clearly separating from the occupation areas of the two former varieties” should be “In contrast, A. sieboldii grows in central and eastern China with the Qinling-Dabashan mountainous areas as the center of its distribution center, clearly separating it from the areas occupied by the two former varieties”

Line 56: “chemical sites of” should be “chemicals of”

line 117: How can they place 157,713 unigenes in a GO category if only 128,934  were annotated? How they can place unannotated genes in a GO category if they don’t know what they are?

Line 131: “provide a precious guidance” should be “will provide useful guidance”

Author Response

Dear Professor,

Thank you for your review and valuable comments on the manuscript “Comparative Transcriptomics Analysis for Gene Mining and Identification of a Cinnamyl alcohol dehydrogenase Involved in Methyleugenol biosynthesis from Asarum sieboldii Miq.”. We have made revisions according to your advice.

1. The authors have compiled a useful dataset. The work seems to be performed competently, and the analysis is appropriate. It would be helpful to explain in the introduction and in their results why they chose to focus on CAD genes, rather than waiting until lines 269 and 270 of the discussion. It also seems that there is excessive speculation about the pathway for asarinin synthesis in the discussion since there is no supporting biochemical evidence from their study.

Many thanks for your nice suggestions. More remarks about the purpose why we chose to focus on CAD genes have been supplemented. In addition, the sentence “it is significant in A. sieboldii to isolate and characterize AsCADs, which are involved in coniferyl alcohol production” added in Introduction.

In Discussion, we simplified the speculation about the pathway for asarinin synthesis as your suggestion. it looks better. 

2. Lines 50-52: “Contrarily, A. sieboldii takes place in central and eastern China with the Qinling-Dabashan mountainous areas as distribution center, clearly separating from the occupation areas of the two former varieties” should be “In contrast, A. sieboldii grows in central and eastern China with the Qinling-Dabashan mountainous areas as the center of its distribution center, clearly separating it from the areas occupied by the two former varieties”

It has been corrected accordingly.

3. Line 56: “chemical sites of” should be “chemicals of”

It has been corrected.

4. line 117: How can they place 157,713 unigenes in a GO category if only 128,934 were annotated? How they can place unannotated genes in a GO category if they don’t know what they are?

The 128,934 annotated ungenes we counted are from four databases including Nr, Swiss-Prot protein, Cluster of Orthologous Groups (COG), and the Kyoto Encyclopedia of Genes and Genomes (KEGG). In a GO category statistic analysis, some unigenes possessed assignment to more than more gene ontology categories. So the number of unigenes annotated on GO categories are more than 128,934.

5. Line 131: “provide a precious guidance” should be “will provide useful guidance”

It has been corrected.

Thank you and best regards.

Yours sincerely,

Zhong Liu, Dongming Ma, Jinjie Liu

Reviewer 3 Report

In this study “Comparative Transcriptomics Analysis for Gene Mining and Identification of a Cinnamyl alcohol dehydrogenase Involved in Methyleugenol biosynthesis from Asarum sieboldii Miq”, Liu et al. showed transcriptomic sequencing and analysis from different tissues at different development stages from Asarum sieboldii focusing on Methyleugenol and Asarinin biosynthesis pathway.

The results of this manuscript are interesting; however, I would suggest revising several points in order to improve it:

The number of different tissues is not clear. The authors investigated on transcriptomic sequencing from five tissues at different development stages (lene88). In section 2.1 (line 101-102) there were only three different kinds of tissues (root, rhizome and leaf). In figure 2B bar-plots showed another tissue, represented by petiole. Please, clarify this aspect.

In order to clarifying the expression of AsCAD genes, the relative expression level was evaluated in different tissues at different stage. In a relative expression generally was used an untreated sample or reference tissue in a particular stage. What did they mean “relative expression”? Relative expression to what?

The authors also studied different developmental stages of different tissues, but there are not the motivations of this experimental approach. Please, clarify this aspect.

The authors should increase the resolution of all figures.

The references have to be carefully checked.

Please, check abbreviations in all article.

Author Response

Dear Professor,

Thank you for your review and valuable comments on the manuscript “Comparative Transcriptomics Analysis for Gene Mining and Identification of a Cinnamyl alcohol dehydrogenase Involved in Methyleugenol biosynthesis from Asarum sieboldii Miq.”. We have made revisions according to your advice.

1. The number of different tissues is not clear. The authors investigated on transcriptomic sequencing from five tissues at different development stages (lene88). In section 2.1 (line 101-102) there were only three different kinds of tissues (root, rhizome and leaf). In figure 2B bar-plots showed another tissue, represented by petiole. Please, clarify this aspect.

For this aspect, it is because the materials used in experiments have not been explained clearly. Actually, the tissues employed throughout in the study were three nutritional organs, i.e,, root, rhizome and leaf. In transcriptome sequencing, the three organs at anthesis (May) were taken into analysis, representing various tissues of the plant. Here, leaf included leaf blade and petiole. Correspondingly, in section 2.1 the “leaf” of the sentence “A morphological feature of the intact plant and the representative morphology of root, leaf and rhizome were shown in Figure 1.” was entire leaf (including both leaf blade and petiole). We have made revision on Line 106-107. Moreover, considering the rhizome to be the main medicinal part of the plant, it was used as representative organ to evaluate variations in different developmental stages. Therefore, rhizomes either at pre-anthesis (March) or at post-anthesis (July) were involved in the sequencing performance. (see 4.1 Plant Material). In a total, there were five tissues applied in the transcriptome sequencing analysis.

Before qRT-PCR experiment, we speculate that petiole possibly contains lignin to some extent. Thus, leaf blade and petiole have been separated as qRT-PCR templates. Now we have revised it as “leaf blade” instead of “leaf”. (see 4.4 Quantitative Real-Time PCR Validation)

2. In order to clarifying the expression of AsCAD genes, the relative expression level was evaluated in different tissues at different stage. In a relative expression generally was used an untreated sample or reference tissue in a particular stage. What did they mean “relative expression”? Relative expression to what?

Previously, relative expression was used to mean the relative expression to 18S rRNA. We revised the Y coordinate as 2-△△Ct and 18S rRNA was used as reference gene, which made the figure more clear.

3. The authors also studied different developmental stages of different tissues, but there are not the motivations of this experimental approach. Please, clarify this aspect.

Because the volatile oil including methyleugenol, the major chemical component of Asarum sieboldii, accumulated to the highest level at anthesis (May), it was expected to elucidate the mechanism of this physiological phenomenon with the help of comparing transcriptomics characteristics between flowering period and other different developmental stages such as pre- or post-anthesis (March and July respectively). We hopefully use more detailed different developmental stages to achieve the candidate genes involved in methyleugenol and asarinin biosynthesis. So, the organ rhizome has been used to investigate the content variation and accumulation principle of volatile oil during different developmental stages. Both root and leaf (including leaf blade and petiole) were taken as comparisons.

4. The authors should increase the resolution of all figures.

We have increased the resolution of all figures.

5. Please, check abbreviations in all article.

We have checked all abbreviations.

Thank you and best regards.

Yours sincerely,

Zhong Liu, Dongming Ma, Jinjie Liu

Reviewer 4 Report

Liu et al conducted transcriptomic sequencing and analysis from different tissues at different development stages from A.sieboldii, and focused on methyleugenol and asarinin biosynthesis pathway. Furthermore, they demonstrated a cinnamyl alcohol dehydrogenase AsCAD5 as the key gene for the formation of coniferyl alcohol. The outline of manuscript is clear and it is a valid piece of work. However, I have the following concerns that the authors should address:

1.       In the end of Abstract, author stated that qRT-PCR results showed AsCAD5 had the highest expression in root, and this result can indicate this gene might have important function in lignin, methyleugenol and asarinin biosynthesis. What is the logic and why did the author could come up with this causality? It is not necessary that more means function.

2.       There are two abbreviations without full names in the Abstract, appearing the first time in the entire manuscript: DIR and FPKM, although they were explained in later text.

3.       I am confused with the number of different tissues that the authors investigated: in page 2, line 88, the authors claimed five tissues; in page 3, line 101-102, from the listed five samples, actually there were only three different kinds of tissues; in Figure 2B, bar-plots only showed four tissues (leaf, petiole, root, and rhizome). How many tissues did the author investigate?

4.       Is there any physical or biological reasons why these genes had the highest expression in root?

5.       The authors also investigated different developmental stages, but seemed not talk much about it. What was the purpose of this part and what can tell us from this part of experiment?

6.       It is not super clear and good to embed tables in figures, e.g. Figure 1B, Figure 7D, I would suggest to separate them as independent tables if the journal does have strict limitation for number of main tables and figures.

7.       The authors applied qRT-PCR to check the gene expression in different organs, what did they mean “relative expression”? Is it not absolute quantification? Relative expression to what? Then does it make sense to compare between different genes? Because the authors claimed that the expression of AsCAD7 is lowest.

8.       Figure 7 A-C: it seems the retention time of coniferyl aldehyde, sinapyl aldehyde, and p-coumaryl aldehyde is exactly the same. Is it true or it was because the authors apply relative retention time as the x-axis. The same for three alcohol standards. Addition, the authors should increase the resolution of figures, it is hard to read.

Author Response

Dear Professor,

Thank you for your review and valuable comments on the manuscript “Comparative Transcriptomics Analysis for Gene Mining and Identification of a Cinnamyl alcohol dehydrogenase Involved in Methyleugenol biosynthesis from Asarum sieboldii Miq.”. We have made revisions according to your advice.

1. In the end of Abstract, author stated that qRT-PCR results showed AsCAD5 had the highest expression in root, and this result can indicate this gene might have important function in lignin, methyleugenol and asarinin biosynthesis. What is the logic and why did the author could come up with this causality? It is not necessary that more means function.

Thanks for your good suggestion with care. We agree to your comments, and we delete the speculation function of the sentence “further suggesting AsCAD5 might function importantly in biosynthesis of lignin, methyleugenol and asarinin”. Indeed, the function of the gene in plant should be done in the future to clarify the actual role in lignin, methyleugenol and asarinin biosynthesis.

2. There are two abbreviations without full names in the Abstract, appearing the first time in the entire manuscript: DIR and FPKM, although they were explained in later text.

The explanation of the two abbreviations has been added in the abstract.

3. I am confused with the number of different tissues that the authors investigated: in page 2, line 88, the authors claimed five tissues; in page 3, line 101-102, from the listed five samples, actually there were only three different kinds of tissues; in Figure 2B, bar-plots only showed four tissues (leaf, petiole, root, and rhizome). How many tissues did the author investigate?

For this aspect, it is because the materials used in experiments have not been explained clearly. Actually, the tissues employed throughout in the study were three nutritional organs, i.e,, root, rhizome and leaf. In transcriptome sequencing, the three organs at anthesis (May) were taken into analysis, representing various parts of the plant. Here leaf included both leaf blade and petiole. Exactly, in section 2.1 the “leaf” of the sentence “A morphological feature of the intact plant and the representative morphology of root, leaf and rhizome were shown in Figure 1A.” actually meant both leaf blade and petiole. We did the revision on Line 106-107. Moreover, the rhizome was used as representative tissue to investigate variations in different developmental stages, in which rhizomes either at pre-anthesis (March) or at post-anthesis (July) were involved. In a total, there were three organs but five samples in transcriptomic sequencing analysis (see 4.1 Plant Material).

Before qRT-PCR experiment, we speculate petiole possibly contains lignin to some extent. Thus, leaf blade and petiole have been further separated as qRT-PCR templates. Now we have revised it to be “leaf blade” instead of “leaf”. There were used totally four samples in this analysis, that is, root, rhizome, leaf blade as well as petiole (see 4.4 Quantitative Real-Time PCR Validation).

4. Is there any physical or biological reasons why these genes had the highest expression in root?

Methyleugenol and asarinin are important secondary metabolism constituents of root in A. sieboldii Miq. CADs have been reported to be involved in lignin biosynthesis from many references. From our experiments, AsCADs, especially AsCAD5, has the biochemical functions in catalyzing p-coumaryl, coniferyl, and sinapyl aldehydes to their corresponding alcohols, which are key intermediates employed not only in biosynthesis of lignin but also in that of methyleugenol and asarinin. We think, this is one of the reasons why these genes had the highest expression in root.

5. The authors also investigated different developmental stages, but seemed not talk much about it. What was the purpose of this part and what can tell us from this part of experiment?

Because the volatile oil including methyleugenol, the major chemical component of Asarum sieboldii, was accumulated to the highest level at anthesis (May) of this plant. We hopefully use more detailed different developmental stages to achieve the candidate genes involved in methyleugenol and asarinin biosynthesis. So, the organ rhizome has been used to elucidate the content variation and accumulation principle of volatile oil during different developmental stages. Both root and leaf (including leaf blade and petiole) were taken as comparisons.

6. It is not super clear and good to embed tables in figures, e.g. Figure 1B, Figure 7D, I would suggest to separate them as independent tables if the journal does have strict limitation for number of main tables and figures.

  Thank you for your suggestion. It has been changed. Now there are totally 9 figures and 3 tables in the manuscript.

7. The authors applied qRT-PCR to check the gene expression in different organs, what did they mean “relative expression”? Is it not absolute quantification? Relative expression to what? Then does it make sense to compare between different genes? Because the authors claimed that the expression of AsCAD7 is lowest.

Previously, relative expression was used to mean the relative expression to 18S rRNA. We revised the Y coordinate as 2-△Ct and 18S rRNA was used as reference gene, which made the figure more clear.

8. Figure 7 A-C: it seems the retention time of coniferyl aldehyde, sinapyl aldehyde, and p-coumaryl aldehyde is exactly the same. Is it true or it was because the authors apply relative retention time as the x-axis. The same for three alcohol standards. Addition, the authors should increase the resolution of figures, it is hard to read.

There are only minor differences among the six chemical standards' retention time. We have modified Figure 9 to make it easily distinguished. In addition, the specific report about their retention time has been supplemented as the zip file “six chemical standards' retention time reports”.

Thank you and best regards.

Yours sincerely,

Zhong Liu, Dongming Ma, Jinjie Liu